# The Glymphatic System May Play a Vital Role in the Pathogenesis of Hepatic Encephalopathy: A Narrative Review

**DOI:** 10.3390/cells12070979

**Published:** 2023-03-23

**Authors:** Ali Sepehrinezhad, Fin Stolze Larsen, Rezan Ashayeri Ahmadabad, Ali Shahbazi, Sajad Sahab Negah

**Affiliations:** 1Department of Neuroscience, Faculty of Advanced Technologies in Medicine, Iran University of Medical Sciences, Tehran 1449614535, Iran; sepehrinezhad.a@iums.ac.ir (A.S.); shahbazi.a@iums.ac.ir (A.S.); 2Neuroscience Research Center, Mashhad University of Medical Sciences, Mashhad 9919191778, Iran; 3Department of Gastroenterology and Hepatology, Rigshospitalet, Copenhagen University Hospital, 999017 Copenhagen, Denmark; fin.stolze.larsen@regionh.dk; 4Shefa Neuroscience Research Center, Khatam Alanbia Hospital, Tehran 1449614535, Iran; re.ashayeri@gmail.com; 5Cellular and Molecular Research Center, Iran University of Medical Sciences, Tehran 1449614535, Iran

**Keywords:** glymphatic system, hepatic encephalopathy, astrocytes, brain fluids, waste substances

## Abstract

Hepatic encephalopathy (HE) is a neurological complication of liver disease resulting in cognitive, psychiatric, and motor symptoms. Although hyperammonemia is a key factor in the pathogenesis of HE, several other factors have recently been discovered. Among these, the impairment of a highly organized perivascular network known as the glymphatic pathway seems to be involved in the progression of some neurological complications due to the accumulation of misfolded proteins and waste substances in the brain interstitial fluids (ISF). The glymphatic system plays an important role in the clearance of brain metabolic derivatives and prevents aggregation of neurotoxic agents in the brain ISF. Impairment of it will result in aggravated accumulation of neurotoxic agents in the brain ISF. This could also be the case in patients with liver failure complicated by HE. Indeed, accumulation of some metabolic by-products and agents such as ammonia, glutamine, glutamate, and aromatic amino acids has been reported in the human brain ISF using microdialysis technique is attributed to worsening of HE and correlates with brain edema. Furthermore, it has been reported that the glymphatic system is impaired in the olfactory bulb, prefrontal cortex, and hippocampus in an experimental model of HE. In this review, we discuss different factors that may affect the function of the glymphatic pathways and how these changes may be involved in HE.

## 1. Introduction

Hepatic encephalopathy (HE) is a neurological and neuropsychiatric complication following acute liver failure (ALF) or chronic liver disease (CLD) [1,2,3]. The exact pathophysiological mechanisms of HE are not fully understood. However, numerous studies have demonstrated that HE has complex pathology at the cellular molecular levels, e.g., astrocyte dysfunction plays a critical role in the pathogenesis of both forms of HE (i.e., ALF and CLD) [4,5,6,7,8,9]. Furthermore, substances including ammonia, glutamine, glutamate, aromatic amino acids, lactate, alanine, bile acids, and neurosteroids accumulate in the brain parenchyma during HE [10,11,12] and are directly implicated in the pathogenesis. Intracellular molecules are normally removed by two major protein degradation systems including the ubiquitin-proteasome system and autophagy in the neural cells [13]. Thereafter, cytosolic proteins and wastes are released into the interstitial space. These waste proteins and metabolic products are released into the cerebrospinal fluid (CSF) and finally are drained away by the glymphatic system [14]. This pathway provides a unidirectional fluid current to transport glucose and other nutrients, but the system is particularly known for the clearance of waste substances from the brain interstitial space [15]. Impairment of the glymphatic system, and the subsequent accumulation of waste soluble proteins has been proposed for the development of neurological disorders, such as Alzheimer’s disease [16,17,18], and is implicated in stroke [19], and traumatic brain injury [20,21]. Recently, researchers have mentioned the disruption of the glymphatic system in an experimental model of HE [22,23]. In this review, we focus on the components of the glymphatic pathway that may be compromised in HE.

## 2. An Overview

The glymphatic system is a system for waste proteins and metabolic products clearance from interstitial fluids (ISF) in the brain parenchyma. The name “glymphatic” derives from the major role of astrocytes, the central nervous system’s most abundant glial cells [24] in this system that operates like a lymphatic system. Its unidirectional flow initiates from Virchow-Robin spaces or periarterial spaces, where CSF flows from the subarachnoid space to the brain tissue along the penetrating arterioles (Figure 1). The driving force for the influx of CSF to the ISF and movement of CSF-ISF into the interstitial space is arterial pulsation [25,26]. Astrocytic end-feet express aquaporin 4 (AQP4) water channels, where astrocytes and endothelial cells constitute barriers between CSF and ISF [27,28]. AQP4 channels are bidirectional and facilitate water flux between blood circulation, periarterial space, interstitial space, and perivenous space. These channels decrease the resistance of CSF flow from the periarterial space into the ISF and facilitate the movement of ISF toward the perivenous space due to the expression on astrocyte processes [29]. CSF-ISF fluid subsequently enters the perivenous space and releases interstitial metabolic agents, soluble proteins, and neurotoxic factors, including lactate, glutamine, glutamate, tau, and amyloid peptides. Finally, waste products are released into the meningeal lymphatic vessels in the dura mater and are transported to the deep cervical lymph nodes (Figure 1) [20]. Enhancing the drainage of CSF from brain tissue ensures the proper functioning of the glymphatic pathway as a powerful driving force for CSF-ISF flow.

## 3. Factors Affecting the Function of the Glymphatic System

The normal function of the glymphatic pathway is to eliminate misfolded proteins completely, maintain water and ions equilibrium, and regulate the CSF, ISF, and intracranial pressure. If any element of the glymphatic pathway is impaired, waste accumulates in the brain ISF. Such impairments occur in the development of neurological disorders. Several factors can affect the glymphatic function in the central nervous system (CNS), such as circadian cycle, body posture, aging, and autonomic innervation. The function of the glymphatic system decreases during wakefulness and becomes better during sleep and under anesthesia conditions [25]. Sleep and anesthesia lead to interstitial space expansion and promote the movement of CSF flow in the perivascular spaces in mice [25]. Another study of rodents showed that the glymphatic system is controlled under circadian rhythms, and AQP4 localization on perivascular astrocyte end-feet is highest in sleepy mice [26]. In a human study using intrathecal injection of gadobutrol as a CSF tracer, glymphatic function was also highest over-night compared to daytime, which indicates the role of sleep for maintenance of an efficient glymphatic pathway [30]. Anesthesia following injection of pentobarbital also improved glymphatic activity in mice [31]. In terms of other factors, the effect of body posture was also evaluated in that latter study. Humans and animals display different body postures at every moment and during sleep and wakefulness. Previous studies have shown that body position affects cerebral blood circulation and intracranial pressure [32,33]. In one study, dynamic-contrast-enhanced magnetic resonance imaging (MRI) and a radioactive CSF tracer revealed that CSF-ISF flow and waste clearance of ISF improved in lateral position in rats [34]. Aging is another factor that has negative effects on the function of the glymphatic system. Aging decreases vascular elasticity and, therefore, reduces the power of arterial pulsation [35,36]. Studies in mice have shown that aging suppresses CSF flow, CSF-ISF flow, and clearance of waste substances [37,38]. Decreasing the levels of collagen type IV and other basement membrane proteins is associated with vascular stiffness and suppression of the amplitude of arterial pulsation, which leads to impairment of CSF influx into the interstitial space [39]. Aging also affects the morphogenesis and cellular behavior of astrocytes, which are fundamental to the structure of glymphatic corridors [40,41]. Furthermore, the distribution of AQP4 throughout the brain was changed with aging [42,43]. The autonomic nervous system (ANS) is another factor that affects the glymphatic system. Activation of the sympathetic nervous system (SNS) can exert negative effects on the glymphatic system through the main neurotransmitter norepinephrine. Reciprocally, activation of the parasympathetic nervous system (PNS) through the vagus nerve positively regulates and improves glymphatic function [44]. Administration of dexmedetomidine, a suppressor of norepinephrine release from locus coeruleus neurons, enhanced glymphatic function in the hippocampus of lightly anesthetized rats [45]. Likewise, intracisternal injection of adrenergic receptor antagonists increased CSF influx into the ISF in conscious mice, as evaluated by a CSF tracer [25]. These studies suggest that norepinephrine and basal tone of the SNS are responsible for suppression of glymphatic transports during wakefulness. Stimulation of the vagus nerve through an implanted device enhanced CSF influx into the ISF and improved CSF-ISF flow in the brain of mice [44].

## 4. Direct Evidence of Glymphatic System Impairment in HE

The first evidence of a dysfunctional glymphatic system in HE was reported by Hadjihambi and her colleagues at University College London [22]. In this study, they measured the glymphatic function in brain parenchyma in bile duct ligation (BDL) rats as chronic liver disease-induced HE model, using dynamic contrast-enhanced MRI and mass-spectroscopy. Decreased penetration of the contrast agent gadolinium indicated reduced CSF-ISF flow in the olfactory bulb and prefrontal cortex of BDL rats. These findings correlated with reduced expression of AQP4 in these brain regions [22]. Thus, they concluded that reduced expression of AQP4 mediated glymphatic impairment and behavioral deficits in HE rats [22]. Furthermore, Jung Hsu et al. [46] demonstrated that the distribution of fluorescent ovalbumin dye as an indicator of glymphatic function was impaired in the brain parenchyma and later in deep cervical lymph nodes in BDL rats compared to the control animals. In addition, the BDL rats displayed microglia activation and neuroinflammation. This study indicated that dysfunctional glymphatic clearance was caused by impaired meningeal lymphatic drainage in HE [46]. A review paper by Claeys et al. [47] explained that impaired glymphatic clearance of waste substances may contribute to the pathophysiology of HE. An editorial letter by Gallina et al. [23] also hypothesized that several abnormalities (i.e., cardiomyopathy-heart failure, pulmonary hypertension, alternation of intracranial pressure, hyperdynamic circulation, and decreased venous outflow) might contribute to the impairment of glymphatic function following liver disease and HE. In this review we describe in detail the links between HE pathogenesis and malfunction of the glymphatic system. The presence of such links suggests areas for future research.

## 5. Factors That May Indirectly Affect the Efficacy of Glymphatic System in Hepatic Encephalopathy

### 5.1. Sleep Disturbances in HE

As mentioned earlier, the sleep–wake cycle can affect the glymphatic system. Sleep improves the function of glymphatic system, while wakefulness reduces it. Sleep disturbance is quite common in patients with liver insufficiency well before HE becomes clinically overt [48,49]. A neuroimaging study suggested that sleep disturbance is a valuable diagnostic sign for subclinical HE in cirrhotic patients [50]. Moreover, decreased sleep quality and daily drowsiness measured by polysomnography and sleep indexes have been reported in HE patients [48,51,52]. Long-lasting sleep abnormalities were also revealed in patients with portosystemic shunt encephalopathy [53]. In an experimental model of this type of HE, disruption of sleep patterns and significant reduction of both rapid eye movement sleep and non-rapid eye movement compared to control rats were shown [54]. Similar findings, such as increased total duration of wakefulness and decreased slow-wave sleep and paradoxical sleep times, have been reported in rats with hyperammonemia [55]. Therefore, manipulating the sleep–wake cycle and correcting sleep disturbances can be an effective effort for the recovery of glymphatic system function and consequently result in ameliorating HE.

### 5.2. Cardiomyopathy, Arterial Hypotension, and Impaired Cerebral Blood Flow in Cirrhosis and HE

Starling forces regulate the fluid flux across the cerebral microvessels, but are severely imbalanced in liver failure, especially if the normal function of the BBB is compromised [56,57]. The main driving forces for CSF influx into the interstitial space and CSF-ISF flow in the brain parenchyma are arterial pulsation induced by the amplitude of the cerebral perfusion pressure (mean arterial pressure (MAP) minus intracranial pressure) [58]. Decreased inotropic and chronotropic properties of heart muscles, reduced peripheral vascular resistance, and reduced systemic blood pressure are commonly seen in patients with insufficient liver function [59,60,61]. Ventricular diastolic dysfunction, left atrial enlargement [62], and both systolic and diastolic cardiac failures [63,64] have been reported in cirrhotic patients. Furthermore, an autopsy study has shown ventricular hypertrophy, cardiac valve calcification, interstitial edema, fibrosis in myocytes, and coronary artery atherosclerosis in cirrhotic patients [65]. Cardiomyopathy as systolic and diastolic abnormalities, inotropic and chronotropic dysfunction, increased plasma levels of myocardial enzymes, myocyte swelling and hypertrophy, and gross abnormalities in cardiomyocytes in histological examinations have also been reported in animal models of cirrhosis and HE [66,67,68,69]. Reduced response to vasoconstrictors and accumulation of circulatory vasodilators, such as nitric oxide, carbon monoxide, and endocannabinoids explain the suppression of systemic vascular resistance in liver failure [70,71,72,73]. In patients with acute liver failure and HE, arterial hypotension is very common [74]. Similar findings have also been shown in the experimental model of HE induced by bile duct ligation in rats, where the systolic blood pressure, measured by tail-cuff, significantly decreased compared to the sham group. Furthermore, injection of thioacetamide as a hepatotoxic agent for induction of HE resulted in a decrease in systolic blood pressure [75]. Dysregulation of cerebral blood flow (CBF) is another important contributing factor that may affect glymphatic transports. Normal regulation of CBF plays a critical role in the supply of vital nutrients to the neuronal cells and elimination of waste agents from brain parenchyma. The cerebral microcirculation is controlled by the partial carbon dioxide tension in the blood, the ANS and cerebral perfusion pressure; i.e., by the CBF autoregulation [76,77]. Numerous human and animal studies have reported that the normal regulation of CBF is impaired in liver failure and HE [12,78,79,80,81,82]. The intracranial pressure increases in severe HE due to astrocyte swelling [83,84,85,86,87,88,89]. Intracranial hypertension also leads to a reduction in cerebral perfusion pressure in patients with acute liver failure [86,90,91,92,93,94]. The combination of arterial hypotension, intracranial hypertension, and impaired regulation of CBF will tend to compromise all the transition pathways of the glymphatic system, aggravating the accumulation of waste metabolic products in the ISF as previously observed in patients with acute liver failure [95,96,97]. This further underscores the importance of the current clinical practice to promptly correct arterial hypotension and hyperammonemia to avoid brain edema and high intracranial pressure, thereby improving CSF distribution in para-arterial spaces and CSF-ISF flow in brain interstitial space.

### 5.3. Alteration in the Function of Autonomic Nervous System in HE

As mentioned in Section 3, ANS affects function of the glymphatic system. Hyperactivity of sympathetic tone and suppression of vagal tone is present in cirrhosis [98,99,100,101]. A recent study showed ANS imbalance in patients with chronic liver disease [102]. Activation of the SNS and elevated levels of norepinephrine in the systemic circulation of cirrhotic patients have been reported [98,103]. Furthermore, PNS activity, as the vagal tone, is strongly suppressed in patients with end-stage liver disease and candidates for liver transplantation [101]. Another study showed an increase in the tonicity of the SNS in patients with high grades of HE compared to healthy control individuals [104]. There was also a considerable change in the function of ANS in patients with HE in another research [105]. Hyperactivity of the SNS is also seen in experimental models of HE induced by thioacetamide in rats [106]. Plasma levels of norepinephrine were found to be increased in rat models of cirrhosis [107,108]. Regarding these findings, modulation of ANS function to restore parasympathetic balance could be of importance to preserve the normal functioning of the glymphatic system, but further study is needed.

### 5.4. Astrocyte Dysfunction and AQP4 Mislocalization in HE

The normal functioning of the glymphatic system in the clearance of ISF requires the presence of healthy astrocytes as a main component. Astrocytes are the most abundant brain glial cells, especially in the cerebral cortex, which regulate ion equilibrium in the extracellular space, provide nutrient agents for their surrounding cells, particularly neuronal cells, release neurotransmitters at the location of astrocyte-neuron and astrocyte-microglia interactions, repair injured regions of CNS, support cerebral microvessels endothelial cells, contribute to the BBB, and finally detoxify many neurotoxic substances (e.g., ammonia and osmotic agents) from the brain interstitial space [10,109,110,111]. The function and morphology of astrocytes are strongly impaired following HE [10]. Glial fibrillary acidic protein (GFAP) is an intermediate filament protein that is intensely expressed in astrocyte cells. GFAP plays an important role in maintaining cell structure, cell proliferation, astrocyte-neuron communication, and in the formation of glial scars following brain injury [112,113]. Postmortem analysis of brain samples from HE patients has revealed decreased protein levels of GFAP in the cerebral cortex and basal ganglia [114]. In the same way, experimental studies on animal models of HE have reported decreased expression of GFAP in several brain areas, such as substantia nigra, ventral tegmental area, hippocampus [115], sensorimotor cortex, thalamus [116], and subcortical white matter [117,118]. In hyperammonemic conditions, astrocytes are primarily the cells that detoxify ammonia from brain parenchyma through their glutamine synthetase (GS) enzyme [119]. The enzyme catalyzes the conversion of ammonia and glutamate to glutamine [119]. Glutamine, which is an active osmolyte, triggers the influx of water into the cell. In hyperammonemic conditions, accumulation of glutamine following ammonia detoxification leads to a hypertonic state, and the influx of water into the astrocytes results in astrocyte swelling [119,120,121,122]. A set of pathological agents and processes, such as inflammation, oxidative stress, ATP depletion, cell senescence, osmolyte accumulation, mitochondrial dysfunction, impaired GFAP, and AQP4 expression triggers and even exacerbates astrocyte swelling following hyperammonemic conditions [10]. Microglia activation-induced neuroinflammation is another factor by which production of proinflammatory cytokines and production of oxidative stress worsens astrocyte swelling following hyperammonemia [123,124,125]. Normal homeostasis of the internal environment of the brain tissue requires an intact BBB in the region of cerebral microvasculature [126]. The BBB is made up of several components: endothelial cells that are joined together with tight junctions, pericytes that are located on the abluminal side and incompletely cover the endothelial cells, and astrocyte end-feet that act as linkers between neuronal networks and blood vessels [127]. This barrier is semipermeable and allows some ions and crucial nutrients (i.e., glucose and amino acids) to transport between blood and brain parenchyma, which ensures the survival of neuronal cells. However, it should be mentioned that the BBB is strongly impermeable to substances for which it does not have a transporter due to tight junctions composed of occludin, claudins, and junctional adhesion molecules; thus, it prevents the entry of circulatory substances into the brain parenchyma [128]. Impairment of this barrier can make it permeable to many circulatory neurotoxic agents, such as ammonia, bile acids, proinflammatory cytokines, and bacterial toxins like lipopolysaccharides [129,130]. Disruption of the BBB may also cause penetration of inflammatory cells such as phagocytic cells, lymphocytes, and antigen-presenting cells into the brain tissue and result in the induction of neuroinflammation and production of reactive oxygen species [131,132,133,134,135]. Therefore, increasing the permeability of the BBB can exacerbate astrocyte damage/swelling, as the main component of glymphatic system, through induction of neuroinflammation and oxidative stress and activation of microglia cells. BBB disruption has been reported in the pathogenesis of HE from experimental to clinical studies. For example, treatment of the human cerebral microvessel endothelial cell (hCMEC/D3) line as an in vitro model of the BBB with plasma samples from cirrhotic patients with HE increases trans-endothelial migration of leukocytes [136]. Increased extravasation of Evans blue dye, as an indicator of BBB disruption, in mice models of HE has been reported [137,138]. Ultrastructural analysis of cerebral cortex, using transmission electron microscopy and [^3^H] inulin, in a rat model of HE has revealed increased leakage in cerebral capillaries [139]. Moreover, the protein level of S100 calcium-binding protein B, as a marker of BBB disruption, increased in serum samples of HE patients [140,141]. As a result, BBB dysfunction in HE is linked with astrocyte swelling and cerebral edema [139,142,143,144,145,146,147,148,149,150,151,152,153,154]. Microglia activation and neuroinflammation are triggered by astrogliosis and induction of the reactive inflammatory A1 phenotype subtype of astrocytes [155]. These reactive phenotypes release pro-inflammatory cytokines such as TNFα and IL-1β which induce BBB disruption, neurodegeneration, and gliopathy [156,157,158,159].

AQP4 water channels are integral membrane proteins that are localized on astrocytic end-feet surrounding the capillaries and facilitate the maintenance of cerebral water flow through their bidirectional properties [160]. However, AQP4 is expressed on plasma membrane instead of perivascular end-feet in swollen and reactive astrocytes [161,162,163,164,165,166]. This AQP4 mislocalization may be one of the main reasons for glymphatic impairment in several neurological diseases (Figure 2) [21,163,167,168]. For instance, the loss of perivascular AQP4 localization as a main part of the glymphatic system was assessed on clearance of amyloid β (Aβ) in human postmortem cases and the α-syntrophin knockout mouse model, which lacks perivascular AQP4 localization. Data indicated that the perivascular localization of AQP4 is significantly decreased in frontal cortical gray matter of AD cases compared to cognitively intact subjects. In the α-syntrophin knockout mouse, CSF influx, CSF-ISF flow, and perivenous interstitial fluids efflux were significantly decreased, while Aβ was increased [167]. Mislocalization of perivascular astrocyte AQP4 and impairment of the glymphatic system have been well identified in AD [16,169,170,171]. Postmortem cerebral cortex analyses from AD patients found that the loss of immunofluorescence for perivascular AQP4 was associated with misfolded amyloid-β [167,172]. The AQP4 channels were also decreased in perivascular reactive astrocytes of the APP/PS1 mice model of AD and aged mice, and the mislocalization was accompanied by a decrease in the transport of a fluorescent tracer as an indicator of lymphatic clearance in brain interstitial space [37,169]. Mislocalization of AQP4 channels on reactive perivascular astrocytes was also reported in mice models of traumatic brain injury (TBI) [173,174]. Moreover, localization of AQP4 on perivascular reactive astrocytes was impaired in hippocampus of TBI mice together with accumulation of hyperphosphorylated tau proteins in this region [163]. Immunofluorescence and tracer tracking techniques demonstrated that perivascular AQP4 mislocalized on reactive astrocytes was associated with a 60% decrease in glymphatic function in TBI mice [21]. Impairment of glymphatic clearance was also reported in the limbic system of TBI rats using dye tracing and MRI [175]. Astrocyte swelling and astrogliosis along with mislocalization of perivascular astrocytic AQP4 as well as glymphatic impairment were also revealed in stroke [162,176,177,178,179]. It is concluded that mislocalization of AQP4 from perivascular end-feet to the cell body in reactive astrocytes is associated with the reduction of efficacy of the glymphatic system and the onset of pathological cerebral findings following AD, TBI, and stroke (Figure 2). This raises the question, does the same mechanism occur in HE? In the context of liver diseases, it seems reasonable to consider similar mechanisms for HE onset. Astrocyte swelling, Alzheimer type II astrocytes, and reactive astrocytes are also predominant cerebral changes in HE that are associated with neurotoxicity [9,180,181,182,183]. These predominant phenotypes of astrocytes may negatively impact glymphatic function in HE as seen in AD, TBI, and stroke. Mislocalization of AQP4 channels in reactive perivascular astrocytes have been previously reported in liver diseases and HE [22,184,185,186,187]. A postmortem analysis of different brain regions (i.e., striatum, cerebellum, thalamus, hippocampus, and cortex) from cirrhotic patients indicated that the expression of AQP4 channels localized to the cell body’s plasma membrane was significantly increased [184]. Furthermore, decreased expression of AQP4 in the olfactory bulb and prefrontal cortex of BDL rats along with the impairment of the glymphatic system, using dynamic contrast-enhanced MRI and mass-spectroscopy, was also observed [22]. In these rats, penetration of gadolinium as an indicator of subarachnoid CSF-ISF flow as visualized by dynamic contrast-enhanced MRI was significantly decreased in prefrontal cortex and olfactory bulbs as compared to the control [22]. Moreover, microarray expression analyses of astrocytes isolated from hyperammonemic mice showed that the expression of the gap junction proteins connexin-43 and AQP4 was decreased compared to the control cells [187]. In addition to that, astrocytes exposed to ammonia in culture medium decreased the expression of AQP4 and changed the arrangement of AQP4 channels on the plasma membrane [186]. Based on the existing data, it is suggested that mislocalization of AQP4 water channels on astrocytes in HE may impair glymphatic clearance and waste aggregation (Figure 2; Table 1.). Further research is needed to confirm the role of the glymphatic system dysfunction in the context of HE.

## 6. Summary

The glymphatic system is an astrocyte-dependent unidirectional transport pathway that empties macromolecules and waste substances from the brain interstitial space. HE mainly evolves and progresses due to hyperammonemia and inflammation [217]. Accumulation of some neurotoxic and osmotic agents such as ammonia, glutamine, reactive oxygen species, glutamate, alanine, aromatic amino acids, lactate, and proinflammatory cytokines, into the brain parenchyma and interstitial space is considered a central issue in the development and progression of HE [10]. Therefore, an efficient glymphatic pathway that can help remove these waste agents from the brain parenchyma may be of potential clinical interest. For instance, it has been shown that an increase in meningeal lymphangiogenesis through injection of adeno-associated virus 8-vascular endothelial growth factor C (AAV8-VEGF-C) into the cisterna magna of bile duct ligation-induced rats with HE reduced microglial activation, improved the glymphatic function, decreased the expression of proinflammatory cytokines in the cerebral cortex, and improved the motor function [46]. However, there remains a paucity of evidence on the impairment of the glymphatic system HE, or on the effect of treatments such as liver transplantation on its function. Further systematic studies are thus warranted. For the first time, Hadjihambi et al. showed evidence of glymphatic system impairment in the BDL rats model of HE using neuroimaging and immunofluorescence techniques [22]. Here we discussed different levels of pathological changes in HE that can be related to the glymphatic pathway, including the sleep–wake cycle, arterial pulsation, astrocyte morphology and function, AQP4 expression patterns, and meningeal lymphangiogenesis, to draw attention to this topic for future studies (Figure 3).

## Figures and Tables

**Figure 1 cells-12-00979-f001:**
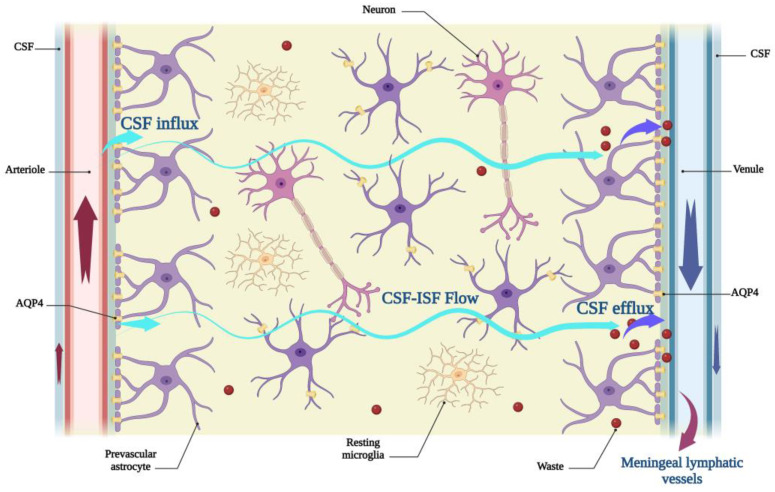
The components of the glymphatic pathway. In the region of Virchow-Robin spaces, CSF along with penetrating arterioles flow into the brain parenchyma. Arterial pulsation provides influx of CSF into the periarterial interstitial space. The system eliminates most neurotoxic agents and waste substances from ISF. AQP4 water channels that are localized in perivascular astrocyte end-feet mediate fluid movements throughout the system. AQP4: aquaporin 4; CSF: cerebrospinal fluid; ISF: interstitial fluid. Created with BioRender.com.

**Figure 2 cells-12-00979-f002:**
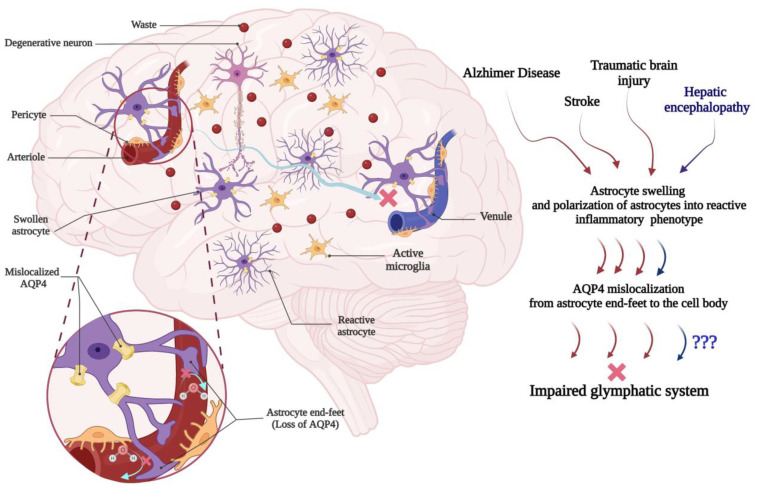
Injured astrocytes differentially express AQP4, impairing the glymphatic system. In several neurological disorders, AQP4 is primarily expressed on the cell body of swollen and reactive astrocytes (A1 phenotype). The AQP4 mislocalization disrupts the flow through the ISF and is responsible for impairment of the glymphatic system, resulting in accumulation of waste metabolites, microglial activation, neuroinflammation, and neuronal toxicity in the brain parenchyma. Similar astrogliosis and mislocalization of AQP4 have been observed in HE. These pathological changes in astrocytes, the main cellular component of the glymphatic pathway, may compromise clearance of ISF in HE. Created with BioRender.com.

**Figure 3 cells-12-00979-f003:**
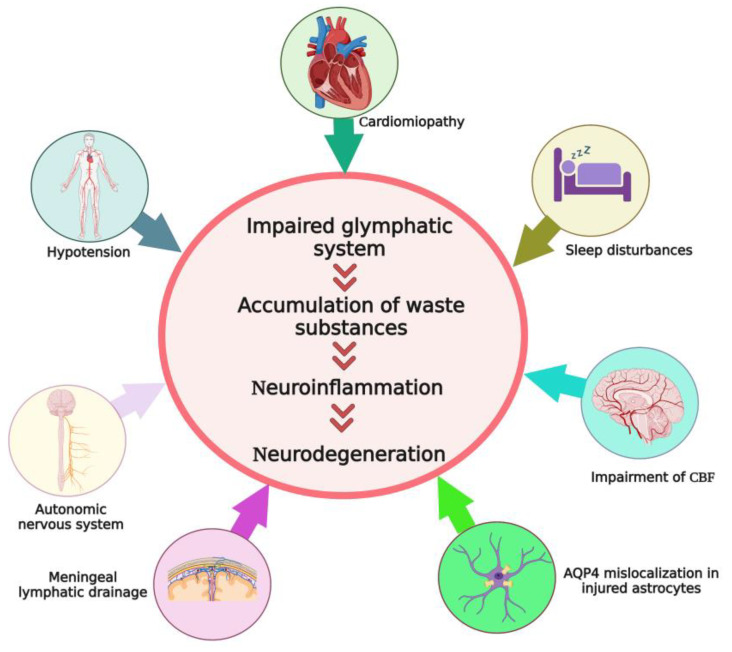
Schematic representations of some pathological processes which may affect normal function of the glymphatic system following HE. All these factors can impair the function of the glymphatic system throughout the cerebral tissue. This induces accumulation of waste substances in the brain interstitial space and triggers neuroinflammation, resulting in neuronal injury and progression of hepatic encephalopathy. CBF: cerebral blood flow; HE: hepatic encephalopathy. Created with BioRender.com.

**Table 1 cells-12-00979-t001:** Main effective factors on the normal function of glymphatic system following liver diseases and HE.

Studies	Affected Factor	Key Results	References
Clinical human cases	Sleep disorders	Sleep disturbances are defined as common features of liver diseases and indicators of subclinical HE in cirrhosis, evaluated by Pittsburgh Sleep Quality Index, Epworth Sleepiness Scale, and polysomnography.	[48,50,53,188,189,190,191]
	Arterial hypotension	Portosystemic shunt and accumulation of vasodilator mediators in systemic circulation, shock, decrease systemic vascular resistance, and blood pressure in HE and cirrhosis.	[70,71,73,74,192,193,194]
	Impaired cerebral blood flow	Decrease in the cerebral perfusion pressure, intracranial hypertension, and impairment of CBF following liver failure and HE.	[83,84,85,86,87,88,89,90,91,92,93,94]
	Altered autonomic nervous system	General imbalance in ANS tone, sympathetic tone hyperactivity, increase in plasma levels of norepinephrine and inhibition of vagal tone in cirrhosis, chronic liver disease and HE.	[98,99,100,101,102,103,104,105]
	Astrocyte dysfunction	Decrease in GFAP expression in the frontal cortex and basal ganglia in postmortem analysis of HE patients, solutes accumulation in astrocytes, brain edema, and astrocyte swelling in patients with fulminant hepatic failure.	[114,144,150,153]
	Dysregulation of AQP4	Postmortem brain analysis of cirrhosis patients indicated an increase in the expression of AQP4 in cell body.	[184]
In Vivo	Sleep disorders	Paradoxical sleep, disruption of sleep patterns, suppression in the duration of rapid eye movement sleep, non-rapid eye movement sleep, and alternation of circadian rhythms in rat models of HE.	[54,55,190,195,196]
	Cardiomyopathy	Cardiac hypertrophy, myocyte swelling, gross abnormalities in cardiomyocytes in histological examinations, increase plasma levels of myocardial enzymes, systolic and diastolic abnormalities, inotropic and chronotropic dysfunction, and QT interval prolongation in rat models of HE and cirrhosis.	[66,67,68,69]
	Arterial hypotension	Decrease in the systemic arterial blood pressure in animal models of HE and liver failure.	[75,197,198,199,200,201,202]
	Impaired cerebral blood flow	Raising intracranial pressure, cerebral hyperemia, vasogenic edema, and impairment of CBF in hyperammonemic HE rats.	[82,201,203,204,205,206]
	Altered autonomic nervous system	Sympathetic tone hyperactivity and increase in circulatory norepinephrine in rats model of cirrhosis and HE.	[106,107,108,202,207]
	Astrocyte dysfunction	Decrease in GFAP expression in substantia nigra, ventral tegmental area, hippocampus, sensorimotor cortex, thalamus, cerebral edema, astrogliosis, and astrocyte swelling in animal models of HE.	[115,116,117,118,142,208,209]
	Dysregulation of AQP4	Decreased expression of AQP4 in olfactory bulb and prefrontal cortex accompanied with glymphatic impairment in BDL rats.	[22]
	Meningeal lymphatic drainage	Improvement in the severity of HE in an experimental rat model following enhanced meningeal lymphatic drainage (requires further experiments)	[46]
In Vitro	Astrocyte dysfunction	Cell swelling and AQP4 rearrangement in ammonia-exposed astrocytes.	[186,210,211,212,213,214,215,216]
	Dysregulation of AQP4	Decreased AQP4 expression in astrocytes isolated from hyperammonemic mice and AQP4 rearrangement in ammonia-exposed astrocytes.	[187]

## Data Availability

Not applicable.

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
