# Peer review of "The Glymphatic System May Play a Vital Role in the Pathogenesis of Hepatic Encephalopathy: A Narrative Review"

_cells, 2023, doi:10.3390/cells12070979_

Round 1

Reviewer 1 Report

This manuscript entitled “The Glymphatic System in the Pathogenesis of Hepatic Encephalopathy: A narrative review discovers another important point of view in the knowledge of the HE pathophysiology, mainly by highlighting the role of the glial cells and sleep disturbance. A sequence of the events should be interesting to be narrated in the first paragraph, in order to better understanding the importance of the role of this glymphatic system, connecting the hepatic encephalopathy with the astrocytes disfunction.

Author Response

12, December, 2022

To: Cells, Editors-in-Chief

Prof. Dr. Cord Brakebusch and Prof. Alexander E. Kalyuzhny

To: Special Issue Editors (Astrocytes in CNS Disorders)

Prof. Dr. Michael Brenner and Prof. Dr. Vladimir Parpura

We wish to express our appreciation to the reviewers for their insightful comments, which have helped us significantly to improve our manuscript. We have revised our paper accordingly and feel that your comments helped clarify and improve our paper. Please find our response (in blue) to the reviewer’s specific comments (in black) below. In the submitted revised manuscript, we have highlighted the revised text in green color.

Reviewer 1

This manuscript entitled “The Glymphatic System in the Pathogenesis of Hepatic Encephalopathy: A narrative review” discovers another important point of view in the knowledge of the HE pathophysiology, mainly by highlighting the role of the glial cells and sleep disturbance. A sequence of the events should be interesting to be narrated in the first paragraph, in order to better understanding the importance of the role of this glymphatic system, connecting the hepatic encephalopathy with the astrocytes disfunction.

Our response: Thank you for your kindly comment.

Sincerely yours,

Dr. Sajad Sahab Negah, Ph.D.

Department of Neuroscience, Faculty of Medicine, Mashhad University of Medical Sciences, Mashhad, Iran. Tel: +98-51-38002473; Email: sahabnegahs@mums.ac.ir

Reviewer 2 Report

The authors generated a very informative review article on the glymphatic system and the pathogenesis of hepatic encephalopathy. The figures and tables are well designed and quite informative, and they also support the discussion in the review article. There are several grammatical and spelling errors scattered throughout the paper that somewhat affect the readability of the current manuscript; therefore, the authors need to carefully proofread the manuscript to correct these English grammatical and spelling errors. There are several subject-verb agreement errors, run-on sentences, and spelling errors, including some typographical errors of some abbreviations in the manuscript. This informative review article is within the scope of the journal and will be of interest to the journal's readership, especially those in the fields of cell biology, neuroscience, neuropathology, neurology, and general medicine.

Author Response

12, December, 2022

To: Cells, Editors-in-Chief

Prof. Dr. Cord Brakebusch and Prof. Alexander E. Kalyuzhny

To: Special Issue Editors (Astrocytes in CNS Disorders)

Prof. Dr. Michael Brenner and Prof. Dr. Vladimir Parpura

We wish to express our appreciation to the reviewers for their insightful comments, which have helped us significantly to improve our manuscript. We have revised our paper accordingly and feel that your comments helped clarify and improve our paper. Please find our response (in blue) to the reviewer’s specific comments (in black) below. In the submitted revised manuscript, we have highlighted the revised text in green color.

Reviewer 2

The authors generated a very informative review article on the glymphatic system and the pathogenesis of hepatic encephalopathy. The figures and tables are well designed and quite informative, and they also support the discussion in the review article.

There are several grammatical and spelling errors scattered throughout the paper that somewhat affect the readability of the current manuscript; therefore, the authors need to carefully proofread the manuscript to correct these English grammatical and spelling errors. There are several subject-verb agreement errors, run-on sentences, and spelling errors, including some typographical errors of some abbreviations in the manuscript.

Our response: Thanks for your comment. We revised all grammatical and syntax errors in the revised version.

This informative review article is within the scope of the journal and will be of interest to the journal's readership, especially those in the fields of cell biology, neuroscience, neuropathology, neurology, and general medicine.

Sincerely yours,

Dr. Sajad Sahab Negah, Ph.D.

Department of Neuroscience, Faculty of Medicine, Mashhad University of Medical Sciences, Mashhad, Iran. Tel: +98-51-38002473; Email: sahabnegahs@mums.ac.ir

Reviewer 3 Report

The review on the influence of hepatic encephalopathy on the glymphatic system and consequently on brain health and disease is rather novel, well researched for references, well structured and written.

there few English writing and spelling mistakes and typos:

Figure 2: to the left bottom part, it should probably say ¨flow¨, but no ¨folw¨

Line 173: it should rather say liver insufficiency and not liver incufficient

Line 247: be of importance to (not the) preserve...

Sentences in line 282-284, and 364/347 should be rephrased, as incomplete, or with missing additional information.

Overall a very interesting review. However, it would be interesting to have also information about clinical improvement of patients after liver transplantation, including improvement of the glymphatic system in those patients.

Author Response

12, December, 2022

To: Cells, Editors-in-Chief

Prof. Dr. Cord Brakebusch and Prof. Alexander E. Kalyuzhny

To: Special Issue Editors (Astrocytes in CNS Disorders)

Prof. Dr. Michael Brenner and Prof. Dr. Vladimir Parpura

We wish to express our appreciation to the reviewers for their insightful comments, which have helped us significantly to improve our manuscript. We have revised our paper accordingly and feel that your comments helped clarify and improve our paper. Please find our response (in blue) to the reviewer’s specific comments (in black) below. In the submitted revised manuscript, we have highlighted the revised text in green color.

Reviewer 3

The review on the influence of hepatic encephalopathy on the glymphatic system and consequently on brain health and disease is rather novel, well researched for references, well structured and written.

there few English writing and spelling mistakes and typos:

Figure 2: to the left bottom part, it should probably say ¨flow¨, but no ¨folw¨

Line 173: it should rather say liver insufficiency and not liver incufficient

Line 247: be of importance to (not the) preserve...

Sentences in line 282-284, and 364/347 should be rephrased, as incomplete, or with missing additional information.

Our response: All comments were revised in the edited manuscript.

Overall a very interesting review. However, it would be interesting to have also information about clinical improvement of patients after liver transplantation, including improvement of the glymphatic system in those patients.

Sincerely yours,

Dr. Sajad Sahab Negah, Ph.D.

Department of Neuroscience, Faculty of Medicine, Mashhad University of Medical Sciences, Mashhad, Iran. Tel: +98-51-38002473; Email: sahabnegahs@mums.ac.ir

Reviewer 4 Report

Reviewer comments for The Glymphatic System in the Pathogenesis of Hepatic Encephalopathy: A Narrative Review

The subject of this review is interesting and highlights there is little known on the glymphatic system in hepatic encephalopathy.

Comments:

1.      The role of a possible impaired glymphatic system in the pathogenesis of hepatic encephalopathy is very speculative. The review lists factors/systems that have been previously described to be involved in the dysregulation of the glymphatic system in other conditions/diseases and then links these factors/systems that have been demonstrated to be involved in HE to finally hypothesize the glymphatic system may be impaired in HE.

2.      Hablitz and Nedergaard J. Neuroscience 2021 have written an excellent review on the glymphatic system. I don’t believe this review adds anything of value aside the speculation of how the glymphatic system may be impaired in HE.

3.      The authors should highlight and summarize what has already been demonstrated in HE in regards to the glymphatic system; refs Hadjihambi 2019 and Gallina 2019. Weiss et al., 2018 (editorial) also merits to be discussed.

4.      HE can arise due to acute liver failure and chronic liver disease. The pathogenesis of each is different. This review mixes both conditions with the majority written on ALF. This review should distinguish between ALF and CLD.

5.      Since HE involves toxin accumulation in the brain, is an impaired glymphatic system detrimental in HE vs other diseases? What about different regions of the brain? This should be discussed.

6.      CSF production is never discussed.

7.      The review is poorly written and very repetitive.

Author Response

In reviewing the comments, we looked at them point-by-point:  

Comments:

  1. The role of a possible impaired glymphatic system in the pathogenesis of hepatic encephalopathy is very speculative. The review lists factors/systems that have been previously described to be involved in the dysregulation of the glymphatic system in other conditions/diseases and then links these factors/systems that have been demonstrated to be involved in HE to finally hypothesize the glymphatic system may be impaired in HE.

Our response: As we know, there are different pathological changes in HE; therefore, we tried to discuss different levels of pathological changes in HE that can be related to the glymphatic pathway (e.g., sleep-wake cycle, autonomic nervous system, arterial pulsation, astrocyte morphology and function, AQP4 expression patterns and meningeal lymphangiogenesis). It must be noticed that there is little evidence for the glymphatic system in HE at both experimental and clinical grades. To show this limitation, we segregated the studies that directly assessed the glymphatic system in the context of HE (it’s been highlighted in the context). On the other hand, the mechanisms underlying the glymphatic system in HE are presently poorly understood, but issues concerning this topic can be considered a priority for future research in the course of HE. For this reason, we outlined and reviewed studies that have shown glymphatic dysfunction in those pathological changes seen in HE, but they are not assessed yet (the heading for this part has been highlighted). Thus, this manuscript presents a perspective insight for future research. 

  1. Hablitz and Nedergaard J. Neuroscience 2021 have written an excellent review on the glymphatic system. I don’t believe this review adds anything of value aside the speculation of how the glymphatic system may be impaired in HE.

Our response: Thanks for sharing. We pored over it and realized that the authors in the mentioned article mainly reviewed the anatomical structure of the glymphatic system and its difference from the lymphatic system. They also discussed how the glymphatic system can modulate neuronal activity. Finally, Hablitz and Nedergaard J. defined that the glymphatic system played a critical role between CNS and other systems. This article is completely different from our manuscript because we just focused on direct evidence on the glymphatic system and HE as well as the possible mechanisms which may enhance the glymphatic dysfunction in the course of HE.

  1. The authors should highlight and summarize what has already been demonstrated in HE in regards to the glymphatic system; refs Hadjihambi 2019 and Gallina 2019. Weiss et al., 2018 (editorial) also merits to be discussed.

We have previously cited these works in the text in several sections. We highlighted these sections with yellow color. However, we also provided more details about the first reported experimental study by Hadjihambi et al. in the revised manuscript.

  1. HE can arise due to acute liver failure and chronic liver disease. The pathogenesis of each is different. This review mixes both conditions with the majority written on ALF. This review should distinguish between ALF and CLD.

Our response: I would completely agree with you, it would be better to discuss the two forms of HE separately. However, it must be mentioned that the evidence is not enough for ALF and CLD related to the glymphatic system. To this point, we had to merge the studies, however, we defined the forms of HE in the revised manuscript. In the last revision, we revised that astrocyte dysfunction is similar in both forms of HE.   

  1. Since HE involves toxin accumulation in the brain, is an impaired glymphatic system detrimental in HE vs other diseases? What about different regions of the brain? This should be discussed.

Our response: This is a serious question. To the best of our knowledge, however, the majority of studies have focused on the prefrontal cortex in the milieu of HE, as summerized in the table below. For this reason, we tried to cover this area of the brain. Nevertheless, different regions of the brain may involve in the glymphatic dysfunction, it can be recommended for future studies. 

Study level

Ref.

Human

https://www.ncbi.nlm.nih.gov/pmc/articles/PMC9062230/

https://www.nature.com/articles/s41598-020-59433-1

https://pubmed.ncbi.nlm.nih.gov/19787808/

https://journals.plos.org/plosone/article?id=10.1371/journal.pone.0037400

https://www.ncbi.nlm.nih.gov/pmc/articles/PMC6870187/

https://www.sciencedirect.com/science/article/abs/pii/S0197018612001222

https://pubmed.ncbi.nlm.nih.gov/1997505/

https://www.frontiersin.org/articles/10.3389/fneur.2019.00033/full

https://link.springer.com/article/10.1007/s11011-019-00457-6

https://aasldpubs.onlinelibrary.wiley.com/doi/abs/10.1002/hep.1840110524

https://link.springer.com/article/10.1007/BF00964814

https://onlinelibrary.wiley.com/doi/abs/10.1111/j.1471-4159.1986.tb13071.x

Experimental

https://pubmed.ncbi.nlm.nih.gov/21371891/

https://www.ncbi.nlm.nih.gov/pmc/articles/PMC7585539/

https://www.sciencedirect.com/science/article/pii/S1567576921004318

https://pubmed.ncbi.nlm.nih.gov/20530233/

https://www.nature.com/articles/s41598-022-22423-6

https://www.sciencedirect.com/science/article/abs/pii/S0166432821000516

https://www.frontiersin.org/articles/10.3389/fnins.2021.678144/full

https://www.sciencedirect.com/science/article/abs/pii/S0041008X19302819

https://www.sciencedirect.com/science/article/pii/S2352345X18300456

https://www.ncbi.nlm.nih.gov/pmc/articles/PMC6190119/

  1. CSF production is never discussed.

Our response: As your suggestion, we added a background information of the CSF production process in the revised version.

  1. The review is poorly written and very repetitive.

We hope the manuscript after careful revisions meet your high standards. It is important to note that the manuscript has been revised several times, so the current format may differ significantly from the original version. Please consider this when checking the comments with the revised text. 

Round 2

Reviewer 4 Report

Very much improved and I accept the current version for publication.